# Capturing colloidal nano- and microplastics with plant-based nanocellulose networks

Ilona Leppänen [1], Timo Lappalainen[1,2], Tia Lohtander[1,3], Christopher Jonkergouw [4], Suvi Arola [1✉] & Tekla Tammelin [1✉]

Microplastics accumulate in various aquatic organisms causing serious health issues, and have raised concerns about human health by entering our food chain. The recovery techniques for the most challenging colloidal fraction are limited, even for analytical purposes. Here we show how a hygroscopic nanocellulose network acts as an ideal capturing material even for the tiniest nanoplastic particles. We reveal that the entrapment of particles from aqueous environment is primarily a result of the network's hygroscopic nature - a feature which is further intensified with the high surface area of nanocellulose. We broaden the understanding of the mechanism for particle capture by investigating the influence of pH and ionic strength on the adsorption behaviour. We determine the nanoplastic binding mechanisms using surface sensitive methods, and interpret the results with the random sequential adsorption (RSA) model. These findings hold potential for the explicit quantification of the colloidal nano- and microplastics from different aqueous environments, and eventually, provide solutions to collect them directly on-site where they are produced.

[1] Biomass processing and products, VTT Technical Research Centre of Finland Ltd., FI-02044 Espoo, Finland. [2] Biomass processing and products, VTT Technical Research Centre of Finland Ltd., FI-40400 Jyväskylä, Finland. [3] Department of Bioproducts and Biosystems, School of Chemical Engineering, Aalto University, Vuorimiehentie 1, 02150 Espoo, Finland. [4] Department of Bioproducts and Biosystems, School of Chemical Engineering, Aalto University, Kemistintie 1, 02150 Espoo, Finland. ✉email: suvi.arola@vtt.fi; tekla.tammelin@vtt.fi

Plastic pollution entering our environment at an increasing rate is a major problem, especially in the marine environment[1–3]. It is estimated that 8.8 million tons of plastic waste ends up in oceans every year[4]. Fragmentation of the plastic debris caused by erosion results in smaller plastic particles, namely secondary micro- and nanoplastics[5,6]. Primary micro- and nanoplastic particles used in i.e. cosmetics can enter the environment directly[5,6]. The terminology for different size ranges of micro- and nanoplastic particles is still under debate. The term microplastic is generally used for microplastics of all sizes (1 μm–5 mm) and there is officially no lower limit to the size of a microplastic. Only recently have scientists introduced the term nanoplastic for particles smaller than a few micrometers[5,7,8]. Some studies have set the upper limit of nanoplastics to 1000 nm and others to 100 nm[9,10]. In our study, we consider the 100 nm particles as nanoplastic particles and the particles ≥1 μm as microplastic particles. Nanoplastics are especially harmful due to their small size (hard to capture, can enter cells), large surface area (capable to bind e.g. toxins), and colloidal nature (limited means for quantification and qualification)[9]. A few studies have analyzed their presence in aquatic animals such as fish and molluscs where they have been found and quantified proving their existence[11]. In addition, model nanoplastic particles have been shown to accumulate on algal cell surfaces[12], to various organs in mussels[13], and to juvenile zebra fish[14,15] affecting their quality of life. A recent study has also shown their presence in the human placenta[16]. An extensive amount of knowledge on the abundance of microplastic particles in different environments is available[17]. Recent efforts to overcome the plastic challenge highlight how genetically engineered enzymes can degrade plastics[18]. This approach could also be a tool for microplastic management. However, very little is known about the existence of nanoplastic particles mainly due to the technical challenges associated with their capture, separation, and analysis leading to a methodological gap in nanoplastic analytics for particles of 1 nm–1 μm[5,8,19]. To date, there is no means to recover nanoplastics from environmental samples for explicit quantification or for the qualitative analysis since existing methods are based on different filtration and elutriation techniques appropriate only for the larger-sized microplastic particles (ø > 50 μm)[20–22]. At best, particle diameters varying from a few microns up to tens of microns can be extracted via density flotation and methods which are based on migration velocity differences[23]. These restrictions leave a blind spot for the recovery, quantification, and qualification of submicron colloidal plastic particles (ø < 1 μm)[5]. Plant-sourced cellulose nanofibrils (CNF) are colloidal level objects with lateral dimensions of 3–10 nm and length up to micrometers. Their water-responsive nature, self-assembly, and other unique properties have only recently unraveled[24]. More specifically, they can effectively recover e.g. gold ions from waste waters[25] and interact with nanoparticles in general[24]. In the realm of nanoscaled materials, besides hydrophilicity and abundance in nature, the assemblies are highly hygroscopic. Strong interactions with water distinguish nanocelluloses from many other nanomaterials with similar properties in terms of large surface area and high aspect ratio[26].

Here we show that nanocellulose networks can be harnessed to capture and quantify even the most challenging fraction of the colloidal plastics. We evidence the capturing ability by following the fluorescence intensity of plastic particles either in a microfluidic set-up or by simply using nanocellulose films as elements to collect the particles from aqueous dispersions. For these studies, we utilized model polystyrene (PS) particles (ø = 1.0 μm and ø = 100 nm, Supplementary Table 1) with anionic and cationic surface charge and well-defined size distributions to reveal the essential mechanism facilitating the capturing efficiency. We prove the versatility of the nanocellulose-based capturing process using larger polyethylene (PE) particles with a much broader size distribution (ø = 38–45 μm, Supplementary Table 1). We illuminate the influence of environmental parameters on particle adsorption behavior by investigating the nanoplastic binding at different pH levels and NaCl concentrations. Finally, with the well-established model PS nanoparticle system, we introduce an interfacial approach, where the particle adsorption data is coupled with image analysis and a random sequential adsorption model, and hence, we are able to quantify the nanoplastic uptake with kinetic information and provide novel methods for nanoparticle detection.

## Results

**Nanocellulose hydrogels trap nano- and microplastic particles.** We followed the particle capturing capacity of native CNF hydrogels[27] in real-time using microfluidic analysis coupled with fluorescent imaging (Fig. 1a, b), which is a straightforward and qualitative concept to evidence the ability of nanocellulose hydrogel to trap nano- and microplastic particles. For this, we utilized fluorescently labeled anionic and cationic polystyrene nanoplastic and microplastic particles abbreviated as PS(ø100 nm) and PS(ø1μm), respectively. The fluorescence intensity increased in the microfluidic traps containing CNF with each cycle of particles of both size classes (Fig. 1c, d, Supplementary Fig. 1, Supplementary Video 1 for positively charged 1 μm particles, Supplementary Fig. 2 for negatively charged particles). Accumulation of the fluorescence intensity over time was as much as 70% higher for the positively charged nanoplastic system when compared to cationic microplastic system (Fig. 1c, d). The same applied to the negatively charged particles although the overall fluorescence accumulation of negatively charged particles (Supplementary Figs. 2 and 3) was lower compared to positively charged particles (Fig. 1c, d, Supplementary Fig. 3). By analyzing the profile of the accumulated fluorescence inside the CNF hydrogel (Supplementary Fig. 3a) we found out that the nanoplastics were able to penetrate deeper into the hydrogel when compared to the microplastics (Supplementary Fig. 3b). Generally, the fluorescence intensity of a single microplastic particle is significantly higher than that of a nanoplastic particle, and thus, our results indicate that CNF hydrogel has a considerably higher capability to capture nanoplastic particles than microplastic particles.

Hygroscopic nanocellulose assemblies display peculiar water transport properties involving capillary action and diffusion[26]. With the aid of water flux, the small nano- and microplastics seem to be conveyed inside the CNF hydrogel network. Moreover, the large surface area of the porous network enhances cohesion facilitating the entrapment of the particles[28]. The negative overall charge of CNF also promotes the accumulation of the positively charged particles, however, it does not prevent the accumulation of the negatively charged particles.

**Capturing nano- and microplastic particles with self-standing nanocellulose films.** We assessed the ability of self-standing nanocellulose films to collect nano- and microplastic particles using fluorescence spectroscopy. This method allows the direct quantification of the number of plastic particles recovered from aqueous dispersions (Supplementary Fig. 4). By using polystyrene nanoplastic and microplastic particles (abbreviated as PS(ø100 nm) and PS(ø1μm)) with either anionic or cationic charge, we attempt to further elucidate the capturing mechanisms, i.e. whether the electrostatic interactions play a role along with the nanocellulose network hygroscopicity. Furthermore, larger anionic polyethylene particles, abbreviated as PE(ø38–45μm), were

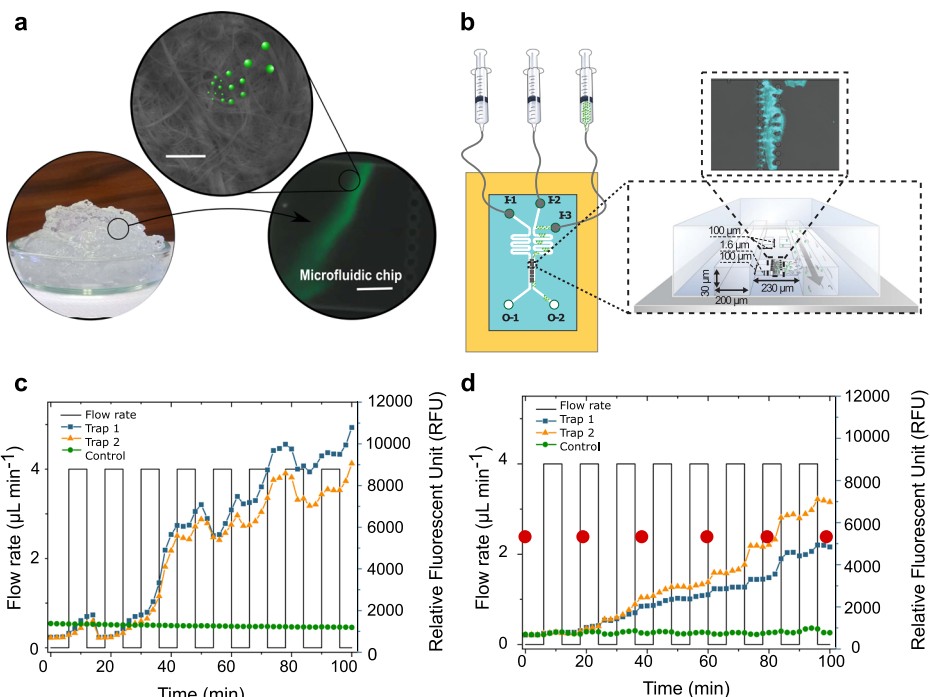

**Fig. 1 Capture of nano- and microplastic particles by native cellulose nanofibril (CNF) hydrogel network. a** Schematic illustration of a proof of concept where the capture of fluorescently labeled polystyrene (PS) nano- and microplastic particles (PS(ø100 nm) and PS(ø1μm)) by CNF hydrogel network is verified using a microfluidic set-up and fluorescent imaging (Supplementary Video 1). Scale bar in the scanning electron microscope (SEM) image is 1 μm and, 25 μm in the microfluidic chip image. **b** Schematic illustration of the microfluidic setup for CNF containing trap showing the injection of fluorescent PS particles (I-3) and water (I-2/I-1). I-1 channel is used to pack the CNF hydrogel into the connected traps and I-2 is used for washing. Fluorescent accumulation of cationic PS(ø100 nm) (**c**) and PS(ø1μm) (**d**) over time by CNF hydrogel network. Green curves show control trap without CNF hydrogel. The orange and blue curves show parallel experiments with CNF in the traps. In (**d**), the red dots indicate the time points where microscopy images were taken (Supplementary Fig. 1). C.J. created the syringes in Fig. 1b using the ChemDraw software, version 20.1.1. from PerkinElmer Informatics. Source data are provided as a Source Data file.

employed to demonstrate that the nanocellulose systems are not particle specific. We used native CNF[27] and TEMPO-oxidized CNF[29] (Supplementary Fig. 5)—the grades with altered water responding tendency due to the phenomenon called hornification[30]. Due to the low anionic charge of native CNF (0.04 mmol g$^{-1}$) it tends to lose its active surface area upon drying whereas the charge of TEMPO-CNF is remarkably higher (1.3 mmol g$^{-1}$) preventing the hornification and, therefore, the water-responsive nature is retained. Polymeric regenerated cellulose (RC) and hydrophobic polystyrene (PS) films were used as reference materials to elaborate the influence of morphology and hydrophobic/hydrophilic balance on the capturing process. CNF grades represent fibrillary nano-porous cellulosic structures, whereas RC with comparable surface wetting behavior is polymeric without distinguishable porosity[31] (Supplementary Fig. 5). Furthermore, the PS surface is assumed to attract PS particles due to hydrophobic effect and similar chemical structure hereby providing a viable reference for CNF. The number of captured polystyrene micro- and nanoplastic particles, (PS(ø100 nm) and PS(ø1 μm)), per unit area of nanocellulose films is shown in Fig. 2a, b, respectively (Supplementary Table 2). SEM images in Fig. 2c show the appearance of the films after being in contact with the nanoplastic dispersion. These results deliver two main messages: (i) highly hygroscopic and anionic TEMPO-CNF film performed the best (Fig. 2a, b) in all cases. Surprisingly, the anionic nanoplastic particles were most efficiently removed by the TEMPO-CNF film. (ii) Attractive electrostatic interactions seem to have a more pronounced effect when dealing with microplastic particles as anionic cellulose films captured cationic microplastics in larger quantities compared to the anionic microplastics (Fig. 2b).

As expected, due to the larger attractive energy between oppositely charged surfaces, the positively charged PS nanoplastic particles are immediately attached to the anionic TEMPO-CNF surface hindering the particle diffusions inside the network, although not fully preventing it. In the anionic system, the attractive energy between negatively charged surfaces is half of that of the oppositely charged surfaces[32], and therefore, anionic nanoplastic particles can enter inside the nanocellulose network more efficiently. Indeed, TEMPO-CNF films captured approximately a third more anionic nanoplastics than cationic nano-plastics. The fact that TEMPO-CNF is able to efficiently capture nanoplastic particles despite the particle charge is a consequence of its nanoscaled porosity coupled with high hygroscopicity enabling peculiar water transport properties. Once the film is in contact with aqueous solutions it swells drastically. TEMPO-CNF network swelling induces capillary flow, which is strong enough to transport the nanoplastic particles into the film network. The large and active surface area of native CNF is partly lost during drying since this CNF grade has a significantly lower surface charge when compared to the TEMPO-CNF. Due to the lack of repulsion between the individual fibrils, upon drying, fibrils aggregate, and severe hornification takes place. Hornification causes irreversible changes in fibril morphology and specific surface area reducing the swelling of the fibril network and thereby lowering the water uptake ability of the system[30]. Thus, hornification explains the lower performance of native CNF films compared to TEMPO-CNF films since never-dried CNF hydro-gels are able to recover micro- and nanoplastics from water flux as discussed above (see Fig. 1). A significantly smaller area of the

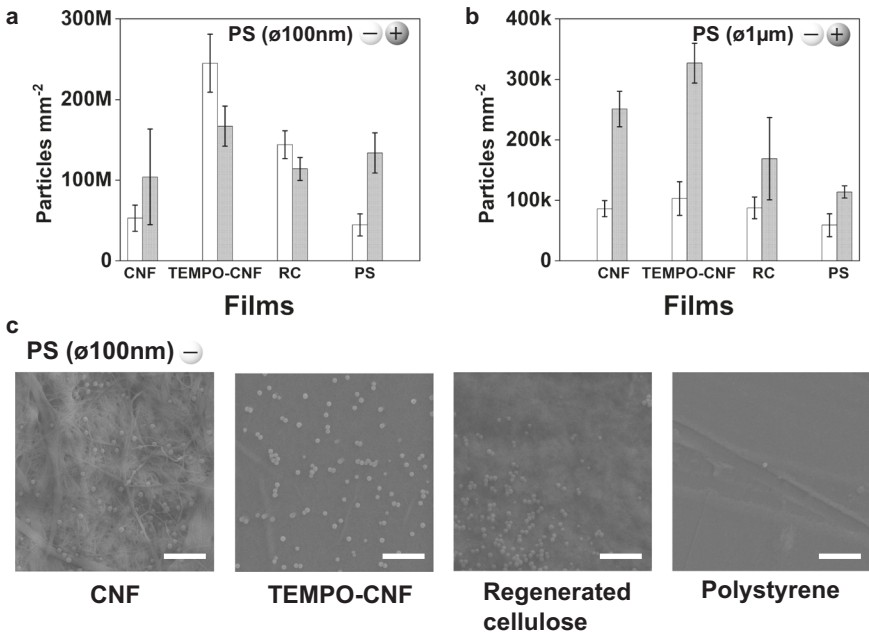

**Fig. 2 Quantitative assessment of entrapped fluorescent nano- and microplastic particles of different size and charge by self-standing films. a** Number of captured nanoplastic polystyrene (PS) particles PS(ø100 nm) and (**b**), microplastic particles PS(ø1μm) calculated based on the fluorescence detection. White bars represent negatively charged plastic particles, and gray bars positively charged plastic particles. The full data for all captured particles are presented in Supplementary Table 2. Error bars in (**a**) and (**b**) indicate mean ± SD. **c** SEM images of the films after being contacted with the anionic PS(ø100 nm) dispersion for 10 min. Scale bar in SEM images is 1 μm. Source data are provided as a Source Data file.

TEMPO-CNF film (30 cm$^2$) is needed to remove all anionic polystyrene nanoplastics from the solution when compared to the native CNF film (140 cm$^2$) (Supplementary Table 3). In the case of polyethylene microplastic particles (PE(ø38–45 μm)), the area needed to recover all PE particles is 16 cm$^2$ and 17 cm$^2$, for CNF and TEMPO-CNF respectively (Supplementary Table 4). Despite the more irregular nature of PE dispersions (large particles with wide size distribution and detergent assisted dispersion stability) both CNF and TEMPO-CNF networks were able to capture PE microplastics from aqueous dispersion. Furthermore, the area needed to recover all PE particles is even smaller than that for PS particles. This clearly evidences the ability of nanocellulose to capture different types of microplastic particles.

For the microplastic particles (PS(ø1μm)), our results showed that all films recover more of the cationic microplastic particles than the anionic ones (Fig. 2b, Supplementary Table 2) indicating a more pronounced role of attractive electrostatic interaction in the capture process compared to the nanosized system (PS(ø100 nm)). Nanocellulose-based systems, however, outperform the polymeric systems (regenerated cellulose (RC) and polystyrene (PS)), due to large surface area, high hygroscopicity, and possible entrapment of particles in the porous network. Lignocellulose-based systems indeed can efficiently trap and transport microplastics from seawater (~1500 plastic pieces kg$^{-1}$ of seagrass) as demonstrated in the seagrass ecosystem[33]. At best, our system—also lignocellulose-derived—can collect roughly 20 billion nanoplastic particles mg$^{-1}$ of TEMPO-CNF, which is a remarkable finding. The ability of regenerated cellulose film to sieve particles cannot be fully explained either by attractive electrostatic interactions, large surface area, nor water interactions and, therefore, we assessed the role of direct surface interactions in the capturing process.

**The role of interfacial interactions—Quantitative method to calculate the adsorption parameters.** To further elaborate the role of surface interactions between nanoplastics and the binding substrates, the adsorption of anionic polystyrene (PS) and polyethylene (PE) particles (Supplementary Table 1) was followed using Quartz crystal microbalance with dissipation monitoring (QCM-D). With this approach, we aim to exclude the influence of network porosity that generates water transport functions and amplify the role of direct surface interactions. We focus our investigations on colloidal-sized well-defined polystyrene particles, (PS(ø100 nm)), since the behavior of nanoscaled particles is mostly taking place at interfaces and the mechanisms are dominated by the interfacial interactions. Furthermore, we briefly tested colloidal PE particles, PE(ø < 450 nm), to widen the understanding of capturing mechanisms occurring at surfaces.

In nature, nanoplastics tend to accumulate e.g. toxins and therefore from the environmental point of view, pure particles do not exist[12]. Therefore, we compared either stabilized or purified polystyrene particles, PS(ø100 nm), all carrying a net negative surface charge (Supplementary Table 1). Our results show that in model conditions (pH 6.8, 10 mM phosphate buffer) PS(ø100 nm) particles—both stabilized and purified—adsorbed on native CNF, regenerated cellulose (RC), and on polystyrene (PS) ($\Delta f_{RC} \gg \Delta f_{CNF} > \Delta f_{PS}$) (Fig. 3a and Supplementary Fig. 6a, c), whereas no adsorption was detected on TEMPO-CNF. This result indicates that the electrostatic repulsion between anionic domains prevents the direct binding of the nanoplastic particles on the highly anionic TEMPO-CNF. Next, the effect of environmental conditions, such as waste and laundry waters were simulated by changing pH and ionic strength of the nanoparticle dispersion. At pH 8 the adsorption behavior was similar to the model conditions (pH 6.8, 10 mM phosphate buffer), whereas slight decrease in adsorption on CNF and RC was detected at pH 10 (Supplementary Fig. 7). No adsorption was detected on TEMPO-CNF. The threshold pH values for wastewaters in Finland are between pH 6–11. However, wastewaters generally have a pH near neutral due to neutralization, which also seems to be the most favorable condition for particle adsorption[34]. Raw laundry detergents tend to have rather high pH values (pH 9–12). However, in use, they

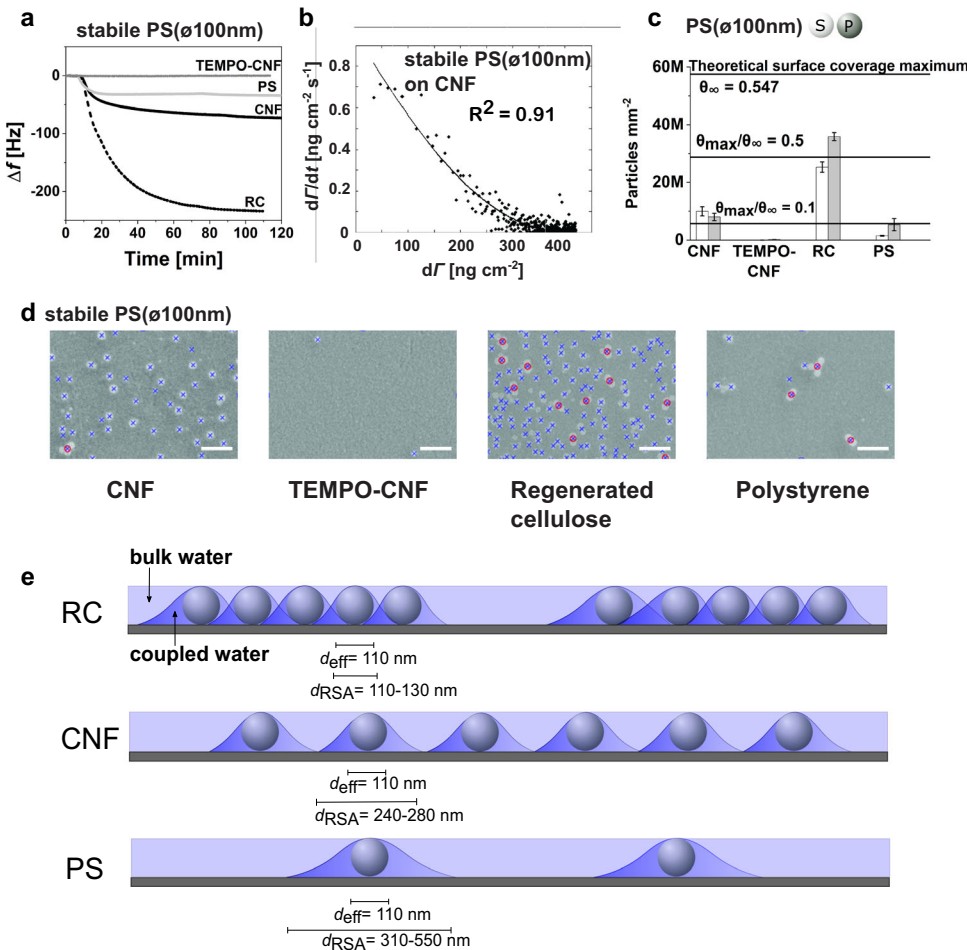

**Fig. 3 Quantitative assessment of surface binding of nanoplastic polystyrene (PS) particles (stabile/purified PS(ø100 nm)) using a surface-sensitive approach, quartz crystal microbalance with dissipation monitoring (QCM-D), coupled with image analysis and fittings with random sequential adsorption (RSA) model. a** QCM-D frequency change responses showing the adsorption of stabile PS(ø100 nm) on TEMPO-CNF (dark gray line), PS (gray line), native CNF (solid black line), and RC (dashed black line). **b** Fitting of the QCM-D adsorption data of stabile PS(100 nm) on CNF using the RSA model. Black dots are measured data, and black line is the RSA fit. The adjusted $R^2$ for the fit is 0.91. **c** Amount of PS(ø100 nm) detected after the adsorption experiments (white bars stabile, gray bars purified) via image analysis of SEM micrographs contrasted to a theoretical surface coverage maximum ($\theta_\infty = 0.547$ which equals to ~5.8 × 10$^{-7}$ circles mm$^{-2}$), which is based on the RSA model. Error bars indicate mean ± SD. **d** Stabile PS(ø100 nm) recognition from scanning electron microscopy (SEM) images using image analysis (Supplementary Fig 12). Blue crosses indicate single particles and red circles indicate identification of clusters. The SEM image scale bar is 0.5 µm. **e** Schematic presentations display the appearance of substrates at the end of the PS(ø100 nm) adsorption process showing the existence of bulk water and water which is strongly interacting with the particles. $d_{eff}$ describes the effective particle diameter, and $d_{RSA}$ is the diameter of the occupied area ($a$) of a single particle including the particle and the coupled water. Source data are provided as a Source Data file.

are strongly diluted with water, which decreases their pH near the pH of tab water (pH 7.3–8.4). It can be concluded that pH variations in these conditions do not change the PS nanoplastic adsorption behavior on CNF or TEMPO-CNF surfaces. With increasing ionic strength (40 mM and 200 mM) the adsorption of colloidal nanoplastic particles increased on both nanocellulose surfaces (Supplementary Fig. 8) indicating that once electrostatic repulsion is efficiently screened, CNF and TEMPO-CNF surfaces are able to bind colloidal PS nanoplastic particles showing the improved capacity to uptake and remove colloidal nanoplastics.

The colloidal fraction of PE particles (PE(ø < 450 nm)) showed no adsorption on any cellulosic surfaces (CNF, TEMPO-CNF, and RC) in model conditions (Supplementary Fig. 9). Analysis of the fractionated sample showed a low amount of PE particles (ø < 450 nm) and zeta potential value of −8.7 mV indicating unstable behavior. These factors together with the density difference of water and PE ($\rho$(PE) < $\rho$(water)) already indicate unfavorable environment for specific surface interactions to take place, and

could tentatively explain why adsorption of PE particles was not detected by QCM-D. However, self-standing films were able to capture PE particles indicating that the main mechanism for entrapment of particles from aqueous dispersion is governed by the high hygroscopicity of the nanocellulose network allowing the particles to be transported inside the structure (Supplementary Table 4).

Finally, we introduce a systematic approach for explicit nanoplastic particle detection to bridge the well-known methodological gap of detection and quantification of nanoplastic particles[19]. We qualified and quantified the substrate performance to bind colloidal plastics in well-defined model conditions (PS(ø100 nm) in pH 6.8, 10 mM phosphate buffer) via surface interactions by comparing the experimental surface coverage to the theoretical maximum coverage. This was carried out by linking the adsorption data to comprehensive image analysis and by applying a random sequential adsorption (RSA) model. In the RSA model, the jamming limit at which the density of adsorbed

**Table 1 Experimental data, adsorption parameters and surface coverage estimations for different nanoplastic particle systems.**

| | Experimental data | | | | | RSA fitting parameters | | | | |
|---|---|---|---|---|---|---|---|---|---|---|
| | QCM $\Delta f_5$ (Hz)[a] | QCM $\Delta D_5$ (×10^{-6})[a] | Coupled water[b] $\frac{\Delta f_5(QCM)}{\Delta f_5(IA)}$ | $\Gamma_{max}$ (ng cm^{-2})[c] | $\theta_{max}$ (solid/air)[d] | $\theta_{max}$ (solid/liquid)[e] | $a \times 10^4$ (nm²)[f] | $d_{RSA}$ (nm)[g] | $k_a \times 10^{-5}$ (cm s^{-1})[h] | $\frac{\theta_{max} \, i}{\theta_\infty}$ |
| CNF + (S) | −75 | 21 | 3.2 | 419 | 0.076 | 0.48 | 6.0 | 280 | 1.0 | 0.14 |
| CNF + (P) | −92 | 24 | 3.1 | 524 | 0.095 | 0.45 | 4.5 | 240 | 1.2 | 0.17 |
| RC + (S) | −230 | 38 | 3.1 | 1330 | 0.24 | 0.35 | 1.4 | 130 | 1.6 | 0.44 |
| RC + (P) | −340 | 47 | 3.2 | 1890 | 0.34 | 0.34 | 1.0 | 110 | 1.9 | 0.62 |
| PS + (S) | −19 | 5.4 | 4.2 | 80.7 | 0.015 | 0.36 | 24 | 550 | 0.2 | 0.027 |
| PS + (P) | −67 | 14 | 4.2 | 281 | 0.051 | 0.36 | 7.6 | 310 | 1.2 | 0.093 |

(S) indicating the stabile PS(ø100 nm) particles and (P) indicating the purified PS(ø100 nm) particles. Source data are provided as a Source Data file.
[a]Changes in frequency and dissipation at the end of the QCM-D measurement after the rinsing step.
[b]Estimation of the amount of water detected at the end of particle adsorption (see Methods).
[c]Maximum surface mass density gained from SEM images and image analysis.
[d]Maximum experimental surface coverage at the end of the adsorption experiment at solid/gas interface determined by image analysis. Particle amount on the surface compared to the theoretical maximum amount calculated based on the RSA model assuming that $\theta_\infty = 0.547$.
[e]Maximum experimental surface coverage at the end of adsorption at solid/liquid interface calculated using Eq. (4), where area $a$ is obtained from RSA fitting.
[f]Occupied area of a single nanoplastic particle including particle and the coupled water i.e. water strongly interacting with the particle.
[g]Diameter of the area ($a$) taken by the particle and coupled water.
[h]Adsorption coefficient describing the affinity of PS(ø100 nm) towards the surface obtained from RSA fitting.
[i]Fractional surface coverage, where $\theta_\infty = 0.547$ is the theoretical maximum surface coverage based on the RSA model. When analyzing the adsorption of purified PS(ø100 nm) on RC substrate, Eq. (3) is valid when $\theta_{max} < 0.3$. Since $\theta_{max}$ for RC is > 0.3, Eq. (5) was applied. If applying the Eq. (6), $d_{RSA}$ would be 96 nm, which is an underestimate since $d_{eff} = 110$ nm.

particles (particles treated as geometrically restricted and fixed circular objects without conformational and orientational changes) saturates on a 2D film is defined as a theoretical maximum coverage ($\theta_\infty = 0.547$). Therefore, the saturation limit in RSA is significantly lower than the optimum filling of the surface[35,36]. By fitting the QCM-D data (Fig. 3a) with the RSA model (Fig. 3b), and by applying image analysis (Fig. 3d, Supplementary Figs. 10–12) we gain access to the adsorption rate and the number of particles per unit area after the adsorption (d$N$/d$t$) (Fig. 3c, Supplementary Table 5, Eq. (2)). This can be translated to surface mass density ($\Gamma$) and adsorption rate (d$\Gamma$/d$t$) (Eq. (6)) since the nanoplastic adsorption process meets well the RSA requirements (Methods). QCM-D detects the adsorbed hydrated total mass by acoustic principle showing simultaneously high changes in energy dissipation (Supplementary Fig. 6b, d, 13). Therefore, the simple Sauerbrey equation (Eq. (1)) cannot be directly used to calculate the mass of the adsorbed particles. To quantify the amount of water in the adsorption process, the areal mass generated from the QCM-D data was rescaled utilizing the surface mass density $\Gamma$ determined from the SEM images by image analysis, which gave the dry mass of adsorbed nanoplastic particles. Figure 3d illustrates the appearance of different substrates after the nanoplastic, stabile PS(ø100 nm), adsorption process and displays the recognition of particles in order to analyze the experimental coverage ($\theta_{max}$) and maximum surface mass density ($\Gamma_{max}$) via image analysis. Table 1 collects the relevant experimental data on particle adsorption, a factor describing water coupled to the adsorbed layer, and the surface coverage ($\theta_{max}$) at the solid-gas interface at the end of the irreversible adsorption process ($t = {\sim}1$ h). Table 1 also tabulates the system-specific parameters from RSA fittings i.e. surface coverage ($\theta_{max}$) at the solid-liquid interface, the adsorption rate coefficient ($k_a$), and the occupied area ($a$) of a single particle including the water, which is strongly interacting with the nanoplastic particle. It should be noted that RSA-derived $\theta_{max}$ takes into account the particle diameter with coupled water resulting in higher surface coverage values when compared to dry systems. Our results show that the strongly bound water layer does not prevent the particle packing and therefore the estimation of the true surface coverage ($\theta_{max}/\theta_\infty$) using dry system data is warranted. The full treatise of the adsorption data with the RSA model is described in the Methods and Supplementary Fig. 14.

We extracted the key findings, and the discussion is supported by the schematic presentation shown in Fig. 3e. (i) Nanoplastic particles had the highest affinity and the highest probability (high $k_a$) to attach on regenerated cellulose suggesting favorable surface interactions positively contributing to adsorption. Particles on regenerated cellulose seem to occupy a smaller area ($a$) allowing a closer packing density indicating that only the synergetic effect of the hydration shell and the electrical double layer of the particles limit the particle packing. Approximately 50% of the theoretical surface coverage maximum was achieved within the time scale of 1 h (Fig. 3c). (ii) Low anionic charge of the native CNF substrate promotes nanoparticle adsorption and direct binding of particles. Approximately 15% of the theoretical surface coverage maximum was achieved with CNF, and the nature of the particle - whether purified or stabilized - had only a minor influence on capturing behavior. (iii) The chemical compatibility and the hydrophobic nature of polystyrene seem not to increase the probability of nanoplastics to adsorb on polystyrene. Nanoplastics had the lowest adsorption probability (low $k_a$) on polystyrene although the probability significantly increased in the purified PS(ø100 nm) system. A circumspect explanation for the high area ($a$) occupied per particle at the solid-liquid interface is originating from the higher amount of coupled water per adsorbed particle indicated by the factor of water being 4 for polystyrene systems (Table 1). The amount of coupled water (the water factor of 3) corresponds to approximately 2/3 of the experimentally sensed total mass for

**Table 2 Nanocellulose hydrogels trap nano- and microplastic particles.**

| Capturing materials | Particles | Key objective | Method |
|---|---|---|---|
| Hydrogel<br>CNF | PS(ø1μm) (+)(−)<br>PS(ø100 nm) (+)(−) | Qualitatively evidence the ability of nanocellulose to trap nano- and microplastics. | Microfluidic set-up coupled with fluorescent imaging |

Main finding: CNF hydrogel captures polystyrene (PS) particles.

**Table 3 Capturing nano- and microplastics with self-standing films.**

| Capturing materials | Particles | Key objective | Method |
|---|---|---|---|
| Self-standing films<br>CNF<br>TEMPO-CNF<br>RC (ref)<br>PS (ref) | PS(ø1μm) (+)(−)<br>PS(ø100 nm) (+)(−)<br>PE(ø38–45μm) | (Semi)-quantitative approach to assess the ability of self-standing nanocellulose films to collect nano- and microplastics.<br>Elucidate the capturing mechanisms, i.e. whether electrostatic interactions play a role along with the nanocellulose network hygroscopicity.<br>Particle specificity | Fluorescence spectroscopy |

Main finding: Hygroscopic TEMPO-CNF film performs the best and attractive electrostatic interactions seem to have a more pronounced role when dealing with ø 1μm particles. Capture of particles is not dependent on the particle type.

**Table 4 The role of interfacial interactions—quantitative method to calculate the adsorption parameters.**

| Capturing materials | Particles | Key objective | Method |
|---|---|---|---|
| Ultrathin films<br>CNF<br>TEMPO-CNF<br>RC (ref)<br>PS (ref) | PS(ø100 nm) (P)(S)<br>PE(ø < 450 nm) | Elaborate role of surface interactions excluding hygroscopicity<br>Influence of environmental conditions (pH, ionic strength) on particle adsorption<br>Direct quantification of adsorbed particles.<br>Tool to assess adsorption kinetics and surface coverage | QCM-D<br><br><br>QCM-D coupled with image analysis and RSA model |

Main findings: Entrapment of particles from aqueous dispersion is mainly governed by the high hygroscopicity of the nanocellulose network allowing the particles to be transported inside the structure. Attractive surface interactions play a role only when the strong repulsive forces are not dominating and the capillary forces are not assisting the capturing process.

CNF and RC surfaces and 3/4 (the water factor of 4) for the PS surface. This is in accordance with other methods[37,38]. Simultaneously, the changes in dissipation for PS systems reach relatively low values suggesting the dominating role of the bulk water over the coupled water in the system.

The adsorption behavior of nanoplastic particles seems to be linked to the amphiphilic nature of cellulose[39], especially when the electrostatic repulsion of the system is relatively weak (native CNF and regenerated cellulose). Cellulosic materials are shown to display different surface properties due to the structural anisotropy[40]. Receding and advancing contact angle values are ranging between 11° and 37° suggesting that cellulose has both hydrophobic and hydrophilic domains[41]. Different wetting behavior of the cellulose surfaces is shown to correlate with the orientation of the crystal planes. The mechanism defining the nanoplastic particle binding is based on the attractive surface interactions only when the strong repulsive forces are not dominating and the capillary forces are not assisting the capturing process.

In summary, we introduce a universal and versatile nanocellulose-based solution, which efficiency to collect and bind micro- and nanoplastic particles are not dependent on any specific physical or chemical interaction. Instead, the entrapment of particles from aqueous dispersion is a result of a dual synergistic feature provided by the nanocellulose network: high hygroscopicity coupled with a high active surface area. These attributes enable the operational assets where colloidal plastic particles—regardless of the size, surface chemistry or plastic type —can be conveyed and captured inside the cellulosic network by exploiting its peculiar water transport properties involving capillary forces and diffusion. Once inside the network, the large

surface area and favorable surface interactions enhance cohesion between the particles and the surface of the material leading to efficient capturing. We have collected the main findings of each experimental set-up with the associated capturing material and particle type to summarize the results (Tables 2–4). We show that by combining surface-sensitive methods with nanomicroscopy, image analysis, and modeling, we are able to quantitatively assess the nanoplastic particle behavior at interfaces. This type of nanoplastic particle adsorption data has not been previously collected, and it is essential when designing materials for quantitative analysis purposes, and for collection and recovery from different environments ranging from wastewaters to the sites where the nano- and microplastics are produced. Nanocellulose originates from the natural sources, it is renewable and non-toxic, which are key aspects when designing next-generation functional materials diminishing the dependency on the synthetic counterparts. Today, nanocellulose can be produced and modified in various ways to yield hydrogels, self-standing films, and porous aerogels and cryogels, which make it an ideal material for many future solutions where the high hygroscopicity is an asset[24,42].

## Methods

**Polystyrene particles.** Fluorescently labeled polystyrene (PS) particles (L9902, L9904, L4655, and L9654 from Sigma Aldrich) of different size (ø = 100 nm and ø = 1 μm) and surface charge (cationic and anionic) were used to analyze micro- and nanoplastic capturing ability, and to reveal the capturing mechanisms of CNF hydrogel and self-standing films. The nanoplastic particles are referred to as anionic (−) /cationic (+) PS(ø100 nm) and the microplastic particles as anionic (−)/cationic (+) PS(ø1μm).

Colloidal nanosized polystyrene (PS) particles (LB1 from Sigma Aldrich) were utilized for QCM-D experiments (ø = 100 nm, i.e. PS(ø100 nm)). These particles were utilized in a *stabile* form (i.e. utilized as provided by the supplier) or *purified*

with the supplier's protocol to remove most of the stabilizing agent. Stabile particles represent the native state of nanoplastics.

Surface charges for all polystyrene particles were determined by zeta potential measurements from 0.1 g L$^{-1}$ dispersions (Zetasizer, Malvern) (Supplementary Table 1).

**Polyethylene particles.** Both fluorescently labeled (ø = 38–45 μm, Cospheric LLC, Santa Barbara, USA, UVPMS-BG-1.025) (PE(ø38–45 μm)) and non-labeled (ø = 200–9900 nm, Cospheric LLC, Santa Barbara, USA, PENS-0.95) (PE(ø200–9900 nm)) polyethylene particles (PE) particles, sizes as reported by the supplier, were utilized. These PE particles were received in powder form and were dispersed in aqueous solution (phosphate buffer pH 6.8, 10 mM) with the aid of a surfactant, Tween20 (0.1 wt%) (Sigma Aldrich) as recommended by the supplier.

For QCM-D studies the non-fluorescent PE particles (PE(ø200–9900 nm)) were filtered to separate the colloidal fraction (Millipore, PVDF membrane, pore size 0.45 μm), which is required for the QCM-D studies (detection range 1 Å–1 μm). The dry weight of the fractionated material (PE(ø < 450 nm)) was measured by drying the material in the oven overnight (105 °C). Dynamic light scattering (DLS) (Zetasizer, Malvern) was used to determine the particle size distribution of the fractionated sample with a concentration of 0.05 g L$^{-1}$. The measurements were performed from three different samples. The size distribution by intensity gave two peaks one at 11 nm ± 0.47 (peak intensity 68%) and the other 450 nm ± 47 (peak intensity 31%) for two of the replicates. The third sample gave an additional small peak at 3200 nm, indicating the formation of clusters. The peak at 11 nm is suspected to be the proprietary additive reported in the Safety Data Sheet of the product as the particles in the dispersion should be ø > 200 nm.

Surface charges for all PE particles were determined by zeta potential measurements (Zetasizer, Malvern) from 0.05 g L$^{-1}$ dispersions (Supplementary Table 1).

**Nanocellulose materials.** Two different grades of cellulose nanofibrils were utilized: mechanically disintegrated cellulose nanofibrils (native CNF)[27] and TEMPO-oxidized (TEMPO = 2,2,6,6-tetramethyl-1-piperidinyloxy radical) cellulose nanofibrils (TEMPO-CNF)[29]. These two nanocellulose materials vary in fibril diameter, surface roughness, and water contact angle (Supplementary Fig. 5). Mechanically disintegrated CNFs were prepared from never dried bleached birch kraft pulp obtained from the Finnish pulp mill. The pulp was first soaked at 1.7 wt% consistency and dispersed using a high shear Diaf dissolver (Minibatch Type20) for 10 min at 700 rpm. The pulp suspension was pre-refined in a Masuko grinder (Supermasscolloider MKZA10-15J, Masuko Sangyo Co., Japan) at 1500 rpm and fluidized with six passes through a Microfluidizer (Microfluidics M-7115-30 Microfluidicis Corp.). The mechanical disintegration resulted in a viscous gel with a final solid content of ~1.6 wt% with an anionic charge of 0.04 mmol g$^{-1}$ analyzed by conductometric titration (SCAN 65:02).

TEMPO-oxidized cellulose nanofibrils (TEMPO-CNF) were produced from bleached softwood kraft pulp obtained from the Finnish pulp mill. Prior to the fibrillation, the pulp was TEMPO-oxidized according to a protocol described by Saito et al.[29], where the never-dried bleached softwood pulp was oxidized with NaClO mediated by 2,2,6,6-tetramethylpiperidine-1-oxyl (TEMPO, Sigma Aldrich). First, the never-dried pulp (cellulose content 60 g) was suspended in 6 L of water and TEMPO (0.94 g) and sodium bromide (6.17 g) were added and the dispersion was mixed for 1 h. The oxidation was started by adding 10% NaClO (11 g) steadily while mixing until the pH reached 10. The pH was maintained at 10–10.2 by adding 1 M NaOH for 2 h. After the reaction had finished the pH was adjusted to 7 with 1 M HCl. The degree of oxidation per anhydroglucose unit was 1.3 mmol g$^{-1}$, determined by conductometric titration (SCAN 65:02). The washed TEMPO-oxidized pulp was subsequently fibrillated with a high-pressure fluidizer (Microfluidics M110P, Microfluidics Int. Co., Newton, MA) with two passes. The final solid content was 1 wt%.

**Microfluidic set-up and fluorescence microscopy.** The capacity of native CNF hydrogel to capture nano- and microplastic particles (fluorescently labeled) was followed using a microfluidic setup and fluorescence microscopy. Microfluidic chips were designed and prepared by standard soft lithography and replica molding approach as previously described[43]. Initially, a master mold including photoresist SU-8 (MicroResist GmbH) on a silicon wafer was created by spin coating two distinct layers, SU-8-5 for the 1.6 μm layer and SU-8-50 for the 20 μm layer. Each layer was exposed to a mercury lamp i-line (3 and 8.5 s, respectively). After development of the topographies, the surface was coated with a ~20 nm fluorocarbon polymer film to facilitate the removal of the PDMS replica after molding[44]. PDMS was prepared by mixing the monomer and crosslinking agent in 10:1 ratio (Sylgard 184 kit, Dow Corning), degassing it, casting it on the microfluidic mold, followed by an overnight curing step at 70 °C. Chips were bonded to glass coverslips by oxygen plasma treatment.

CNF dispersion of 0.5 wt% was prepared for the microfluidic experiments. Before loading the microfluidic channels, the 0.5 wt% CNF solution was centrifuged (14,000 g) for 5 min to spin down large fibrils. The supernatant was loaded into the microfluidic chips with a 500 μl syringe through a single channel, with a flow rate of 300 μm s$^{-1}$. After the CNF fibrils entangled behind the pillars to

form a solid transparent membrane, the CNF solution was switched to H$_2$O. Phase-contrast and fluorescent images were acquired using an Axio Observer Z1 microscope (Carl Zeiss, Jena, Germany)[45]. Images were acquired every 2 min at ×20 magnification during the experiment. The fluorescent signal was obtained from the fluorescent 100 nm and 1.0 μm particles (PS(ø100 nm) and PS(ø1μm)) using excitation light at 480 nm, while collecting the emitted light from 515 to 535 nm. The flow was continually switching between the wash solution (Channel B) and the plastic particle solution (Channel C) every 6 min.

A cross-section profile fluorescence intensity analysis of the CNF hydrogel network was performed on 4 individual traps for each condition in order to gain understanding of the penetration of nano- and microplastic particles (PS(ø100 nm) and PS(ø1 μm)) into the CNF hydrogel network. Cross-sections were acquired in the middle of the washing cycle at 86 min, to ensure only entrapped and bound particles were considered in the analysis. Fluorescence reads were scaled against its peak fluorescence, after which the area under the curve was calculated (Δx 400 nm) and plotted against the distance of the trapped PS(ø100 nm) and PS(ø1 μm).

**Preparation of self-standing films.** Native CNF films were prepared from 0.8 wt% gel, which was cast on polystyrene Petri dishes (ø 9 cm) and dried under ambient conditions for 24 h. The formed CNF films were separated from the plastic supports for further experiments.

TEMPO-CNF films were prepared from 0.2 wt% gel, which were cross-linked with polyvinyl alcohol (PVA, Mowiol 56–98, M$_w$ 195 000 g mol$^{-1}$) according to previously described procedure[46] to enhance the films' wet strength. Briefly, the TEMPO-CNF gel (1 wt%) was diluted to 0.2 wt% and was mixed PVA. The PVA amount used in the films was 10% of the dry amount of TEMPO-CNF. TEMPO-CNF/PVA gel was cast on polystyrene Petri dishes (ø 9 cm) and dried under ambient conditions for 24 h. The formed TEMPO-CNF films were separated from the plastic supports for further experiments. The weight of 1.5 cm × 1.5 cm film was on average 0.0047 g.

Regenerated cellulose (RC) films were prepared by dissolving microcrystalline cellulose powder from spruce (Fluka) in ionic liquid (EMIM[OAc], IoLiTec GmbH) to a 10 wt% solution under heat and mixing (80 °C, overnight). Subsequently, a film was cast on a glass surface using a 510 Coatmaster film applicator (ERICHSEN GmbH & Co. KG, Hemer, Germany) with a gap of 400 μm and speed of 40 mm s$^{-1}$. The regeneration was carried out by immersing the cellulose films in water for 1.5 h. Finally, the regenerated cellulose films were placed between absorbent papers and air-dried at ambient conditions for 3 days.

Polystyrene (PS) films were prepared by dissolving PS pellets (Mw 192,000 g mol$^{-1}$, Sigma Aldrich) in toluene to a 10 wt% solution at ambient conditions overnight. The PS solution was cast on a glass Petri dish and air-dried at ambient conditions for 12 h. The formed PS films were separated from the glass support for further experiments.

**Fluorescence spectroscopy.** Quantification of fluorescently labeled anionic and cationic PS particles captured by self-standing films was conducted using a Cary Eclipse fluorescence spectrophotometer (Varian Scientific Instruments, CA, USA). Fluorescently labeled particles were dispersed to a final concentration of 0.1 g L$^{-1}$ in phosphate buffer (10 mM, pH 6.8). The calibration curves were recorded from five concentrations (0, 0.025, 0.05, 0.075, and 0.1 g L$^{-1}$) for each fluorescent particle at their specific emission maxima (Supplementary Fig. 15). Fluorescence studies were carried out by immersing the films in the aqueous dispersion containing 0.1 g L$^{-1}$ particles for 10 min without mixing (Supplementary Fig. 4). The fluorescence of the solutions was recorded before and after the immersion. The number of recovered particles was calculated by subtracting the fluorescence value after immersion from the fluorescence before the film immersion. All measurements were performed in triplicate with three readings each.

The number of PS particles captured by the films equals the total mass recovered ($m_{Tot}$) divided by the single-particle mass ($m_{particle}$). $m_{Tot} = cV$, where $c$ is the measured concentration, and $V$ is the known volume. The mass of a single PS particle ($m_{particle} = 5.49 \times 10^{-7}$ ng) was calculated, assuming it to be a sphere with a density of $\rho = 1.05$ g cm$^{-3}$ ($\rho$ of PS particles is provided by the supplier).

Quantification of fluorescently labeled PE particles (PE(ø38–45μm)) captured by self-standing films was determined in the same manner as for fluorescently labeled PS particles. The number of particles could not be quantified as for PS particles, since the PE particle dispersion is heterogeneous with respect to the particle size distribution. Thus, the amount of captured PE particles was calculated simply as mass captured by the films by measuring the change in fluorescence before and after immersion, which is directly proportional to the concentration and mass recovered. All measurements were performed in triplicate with three readings each.

**Preparation of ultrathin films for adsorption investigations.** Native CNF, TEMPO-CNF, RC, and PS ultrathin films were prepared by spin coating (Model WS-400BZ-6NPP/LITE, Laurell Technologies, North Wales, PA, USA) the materials on QCM-D sensor crystals (AT-cut quartz crystals with Au or SiO$_2$ surfaces supplied by Q-Sense, Gothenburg, Sweden). The crystals were rinsed with Milli-Q water, dried with nitrogen gas and placed in a UV-ozonizer (Bioforce Nanosciences, CA) for 10 min to clean the surfaces. Prior to CNF or TEMPO-CNF

deposition, a layer of anchoring polymer (polyethylene imine (PEI), 30 wt%, Mw 50,000–100,000 g mol$^{-1}$, Polysciences Inc.) was adsorbed onto the crystal surface by immersing the crystal in 1 g L$^{-1}$ PEI solution for 30 min. The excess of PEI was rinsed away with large amounts of Milli-Q water, followed by nitrogen gas drying.

Transparent dispersion of CNF for spin coating was prepared as described by Ahola et al.[47] Briefly, the CNF gel (1.6 wt%) was diluted to ~0.17 wt% using Milli-Q water and was ultrasonicated (400 W tip sonicator, Branson 450 Digital Sonifier, Branson Ultrasonics, Danbury, USA) for 10 min with 25% amplitude. Subsequently, larger fibril bundles were removed by centrifugation (14,000 g, 45 min). The supernatant with the individual fibrils was collected by pipetting. The Au-coated sensor surface with PEI was first wetted by spin coating 100 μl of Milli-Q water on the sensor at 3000 rpm for 10 s. Subsequently, the individualized nanofibrils were spin coated (3000 rpm, 90 s) on the QCM-D sensor surfaces. After spin coating, the surfaces were rinsed with water, dried gently with nitrogen gas and placed in an oven for heat treatment (80 °C, 10 min).

Ultrathin films of TEMPO-CNF were prepared with a protocol described by Hakalahti et al.[26] The TEMPO-CNF gel (1 wt%) was first diluted to 0.15 wt% using Milli-Q water and subsequently ultrasonicated for 2 min with 25% amplitude to break down aggregates. The TEMPO-CNF surfaces were then prepared from the dilution. Before spin coating of TEMPO-CNF the Au-coated sensor surface with a thin PEI layer was wetted by spin coating 100 μl of Milli-Q water on the sensor at 3000 rpm for 10 s. Subsequently, the nanofibrils were spin coated (3000 rpm, 90 s) onto the QCM-D crystals. After spin coating, the surfaces were rinsed with water, dried gently with nitrogen gas, and placed in an oven for heat-treatment (80 °C, 10 min).

RC surfaces were prepared from trimethylsilyl cellulose (TMSC) ultrathin films by desilylation. TMSC was synthesized by silylation of cellulose powder with hexamethyl disilazane (HMDS), as described previously[48] Two grams of cellulose powder were dissolved in lithium chloride in DMAc. After dissolution the solution was heated to 80 °C and 20 ml of HMDS was added slowly within 1 h. The mixture was cooled and methanol was added in low quantities to enhance the crystallization of TMSC, which proceeded overnight. The TMSC was then filtered and dissolved in tetrahydrofuran and recrystallized in 1000 ml methanol. The methanol washing was repeated a few times and then dried in a vacuum desiccator. The dried TMSC was dissolved into toluene to form a 10 g L$^{-1}$ solution. Prior to spin coating the TMSC, the surface of the SiO$_2$-coated sensor surface was wetted by applying 5 droplets of toluene and spun with the speed of 3000 rpm for 15 s. The TMSC solution was subsequently spin coated onto the SiO$_2$-sensor (3000 rpm, 60 s). The solvent was evaporated by placing the crystals in the oven (60 °C, 10 min). The TMSC ultrathin films were regenerated back to cellulose by desilylation with hydrochloric acid vapor, producing ultrathin RC films.

PS ultrathin films were prepared from a polystyrene solution (0.1 wt% polystyrene in toluene) by spin coating on a gold QCM-D crystal (4000 rpm, 30 s). The solvent was evaporated in the oven (60 °C, 10 min). All coated QCM-D crystals were stored in desiccators.

**Atomic force microscopy (AFM).** Sufficient coverage of the ultrathin films on QCM-D sensors was verified by AFM using a NanoTA AFM + instrument (Anasys Instruments, Bruker, MA, USA) with Mounted Standard Silicon Tapping Mode Probes with Al Reflex coating (Applied Nanostructures Inc., CA, USA). Images of the ultrathin film surfaces were recorded in tapping mode in the air with a scan rate of 0.5 Hz with silicon cantilevers. The damping ratio was around 0.7–0.85 Hz. For each sample, three different areas were analyzed, and the images were not processed by any other means except flattening (Analysis Studio 3.11). AFM images of all ultrathin film surfaces and their height profiles are presented in Supplementary Fig. 5.

**Water contact angles (WCA).** WCAs for ultrathin films were determined to assess the films' surface wettability. We used a sessile drop method with a video camera-based fully computer-controlled contact angle meter (Attension Theta Optical Tensiometer, Biolin Scientific, Finland). A droplet volume of 2 μl (Milli-Q water) and a recording time of 120 s were used to measure the contact angle of the ultrathin films. 2–3 droplets were applied on the ultrathin film surfaces, and the average contact angles were calculated from these. The values used for the calculations were from time point 1.4 s. The value was taken from the beginning of the measurement since the droplet is affected by evaporation due to its small size. WCA values are shown in Supplementary Fig. 5 for all ultrathin films on the upper part of the AFM image.

**Quartz crystal microbalance with dissipation (QCM-D).** Adsorption of PS particles (stabile/purified PS(ø100 nm)) on native CNF, TEMPO-CNF, RC, and PS ultrathin films was investigated using E4 QCM-D instrument (Q-Sense AB, Gothenburg, Sweden). QCM-D is used for following in situ changes of mass at solid/gas and solid/liquid interface, since the measured change in frequency (Δf) corresponds to the change in areal mass (Δm). Simultaneously the change in dissipation (ΔD) is monitored yielding information on the changes in viscoelastic properties of the adsorbed layer. The interpretation of QCM-D data is described elsewhere in detail[49] If the adsorbed film is evenly distributed and rigid, the change in frequency is directly proportional to the change in areal mass and can be calculated according to the Sauerbrey relation presented in Eq. (1)[50]

$$\Delta m = -\frac{C}{n}\Delta f \qquad (1)$$

where Δm is the areal mass, n is the overtone number (n = 1, 3, 5, 7, 9, 11), and C = 17.7 ng (cm$^{-2}$ Hz$^{-1}$) for the 5 MHz AT-cut crystal. Changes in dissipation must remain low (<10$^{-5}$) for the Sauerbrey equation to remain valid. Larger changes indicate a softer and thicker layer, where the amount of adsorbed water is significant.

PS particle adsorption experiments were carried out as follows. Purified and stabile 0.1 g L$^{-1}$ PS particle (ø = 100 nm, PS(ø100 nm)) dispersions were prepared in phosphate buffer (10 mM, pH 6.8). QCM-D sensor surfaces coated with ultrathin films were stabilized in the buffered conditions for at least 12 h. Before introducing PS particle dispersion to the QCM-D chamber, the sensor surfaces were contacted with the phosphate buffer solution for approximately 1 h to avoid the bulk effect. Then 0.1 g L$^{-1}$ PS particle dispersion was introduced into the QCM-D chamber with the flow rate of 0.1 ml min$^{-1}$ for approximately 1 h. The particle adsorption was confirmed by rinsing the surface with phosphate buffer for 1 h. Two replicates of each measurement were performed. The adsorption of PS particles, as well as the possible desorption due to rinsing were monitored by following the changes in frequency (Δf) and dissipation (ΔD) as a function of time.

PS particle adsorption experiments were done in different pH and salt concentrations to simulate different environments. For salt concentration studies, stabile 0.1 g L$^{-1}$ PS particle (ø = 100 nm, PS(ø100 nm)) dispersions were prepared in phosphate buffer (10 mM, pH 6.8) with the addition of either 40 mM NaCl or 200 mM NaCl. For pH studies, stabile 0.1 g L$^{-1}$ PS particle (ø = 100 nm, PS(ø100 nm)) dispersions were prepared in phosphate buffer (10 mM, pH 8) and Borax-NaOH buffer (10 mM, pH 10). The QCM-D measurements were carried out similarly as reported previously for the polystyrene particles (flow rate 0.1 ml min$^{-1}$, particle concentration 0.1 g L$^{-1}$), except for CNF and TEMPO-CNF in 200 mM NaCl the flow rate was lower and thus the adsorption time was recorded for a longer time.

Fractionated polyethylene particles (PE(ø < 450 nm)) were used for the QCM-D studies to further clarify the capturing mechanism and to briefly demonstrate that our system is not particle dependent. The QCM-D measurements were carried in phosphate buffer (pH 6.8, 10 mM, 0.1 wt% Tween20) in the same manner as reported for polystyrene particles (flow rate 0.1 ml/min, particle concentration 0.1 g L$^{-1}$).

**Scanning electron microscopy (SEM) and image analysis.** Self-standing and ultrathin films were imaged after fluorescence and QCM-D measurements with a Merlin Field Emission (FE)-SEM (Carl Zeiss NTS GmbH, Germany) to visualize the films after particle capture and adsorption. The self-standing films and QCM-D crystals were dried after the measurements and attached to SEM sample holders using carbon tape. Samples on the holders were coated with gold by sputtering (2 nm thick gold surface) to improve sample conductivity. Samples were imaged with the electron gun voltage of 3–5 kV and the grid current of 60 pA. The number of pixels in the SEM image was 2048 (H) × 1536 (V), with 256 gray levels. At least three SEM images of each sample were acquired at different positions. In addition, the number of adsorbed PS nanoparticles (stabile/purified PS(ø100 nm)) on the ultrathin films was quantified using image analysis (Matlab), which was developed to recognize the PS nanoplastic particles in SEM images to determine the particle amount per mm$^2$.

In SEM imaging, as in imaging methods general, images are clipped within a rectangular boundary. When a spatial pattern is observed through a rectangular clipping window, several edge defects arise. One of these edge defects is size-dependent sampling bias. Miles has discussed plus-sampling (any object that intersects the clipping window is accepted) and minus-sampling (only those objects that lie within the clipping window are accepted)[51] In our research, no attempt was made to determine the particle size distribution as the particles in the image were the same size. Therefore, size-dependent sampling bias was not a problem in our analysis. The relationship between the actual dimensions of the particles (μm) and the pixel size of the particles was obtained from the scale bar in the SEM image. The SEM images were of good quality; the background variation was small and bright objects (particles) stood out clearly from the dark background. Therefore, no image preprocessing was required, and the first image processing operation was segmentation. It was possible to use a global threshold value because the background variation was small. The thresholding method used in this study was based on histogram shape information. The threshold was chosen for the descending part of the prominent peak of the histogram (see Supplementary Fig. 12). The aim was to identify the individual PS nanoplastic particles and their centers, making it possible to determine the total number of particles. In most SEM images, the particles were detached from each other or formed only small clusters. However, in some cases, the particles had a strong tendency to form clusters (Examples of SEM images with identified particles shown in Supplementary Fig. 11). Thus, the next step of the image analysis was divided into two methods depending on which of the above categories the image was classified into. When the particles were mainly detached, the particles were identified from the threshold image by their area (we know the diameter of particle size in each image) and shape (circular objects). The size of the clusters observed was assumed to be three particles. In samples where the clusters were large, and the particles were mainly in

the clusters, individual particles were not reliably identified. In this case, the area of each cluster was determined, and the number of particles needed to achieve the same area was calculated. Finally, the identified particles were presented by drawing a marker on the original SEM image (see Supplementary Figs. 11, 12).

**Theoretical maximum adsorption of particles—fittings with RSA model**. We used the random sequential adsorption (RSA) model to evaluate the maximum adsorption capacity of the ultrathin films. The thickness of the ultrathin film is well below 100 nm. Thus, the PS particles cannot penetrate the film. If the goal is to determine the maximum number of PS particles that can fit on the surface of ultrathin films, the question can be simplified to the packing of circular disks in a plane. Adsorption of particles on solid, flat surfaces is often an irreversible process, as was also verified by QCM-D measurements in this study (Supplementary Fig. 6). In addition, particles usually do not adsorb one on top of each other, instead they form a monolayer (Supplementary Fig. 10). The basic RSA model assumes that only steric repulsion is present between the circular disks. For circular disks of the same size, saturation occurs at a surface coverage $\theta_\infty$ of 0.547. If there are disks of different sizes (particles of varying diameter) in the system, higher surface coverage $\theta$ can be reached. In this study, all particles were the same size. If the viewing area is one mm² and the PS particles are the same size, the maximum area covered by the particles is 0.547 mm². In this case, a 1 mm² area can hold $5.76 \times 10^7$ circles (PS particles) with an effective diameter (diameter that perceives also the estimation of electrical double layer and hydration shell of the particle) of $d_{eff} = 1.1 d_{abs} = 110$ nm[52]. The cross-sectional area of one particle was calculated using $A_{particle} = \pi(d_{eff}/2)^2$. Also the RSA model assumes that particles hit the surface at the same rate throughout the adsorption process[53]. Therefore, the concentration $c$ must be high enough to form a monolayer in the saturation regime. If there are not enough particles, the adsorption process may stop before reaching the saturation surface coverage. Table 1 shows that the maximum surface mass density ($\Gamma_{max}$) was 1890 ng cm⁻² for purified PS(ø100 nm) adsorbed on RC. With a QCM-D sensor diameter of 9 mm, there was ~1200 ng of purified PS(ø100 nm) on the sensor surface. This corresponds to approximately 0.2% of PS(ø100 nm) (100,000 ng cm⁻³ PS(ø100 nm) dispersion was introduced into the QCM-D chamber with a flow rate of 0.1 ml min⁻¹ for approximately 1 h). Therefore, we can assume that the nanoplastic particles hit the surface at the same rate throughout the adsorption process, and the requirements to utilize the RSA model are met. In our system, all of the main RSA principles are valid, and therefore, the adsorption behavior of the nanoplastic particles can be described using the random sequential adsorption (RSA) model[36].

**Assessment of particle adsorption kinetics and the amount of coupled water**. In order to evaluate the kinetics of the PS particle adsorption process, we modeled the QCM-D data (Fig. 3a, Supplementary Fig. 6) with the RSA model (Fig. 3b, Supplementary Fig. 14) with input value for number of particles gained from the image analysis. The kinetics of adsorption of particles can be described by Eq. (2)[53–55].

$$\frac{dN(t)}{dt} = k_a c' B(\theta) - k_d \theta \qquad (2)$$

where $N(t)$ is the number density of adsorbed particles (in units of 1 cm⁻²), $\theta$ is the surface coverage $\theta = [0,1]$, $t$ is the adsorption time (s), $k_a$ is the adsorption rate coefficient (cm s⁻¹), $k_d$ is the desorption coefficient (cm s⁻¹), $B(\theta)$ is the surface blocking function, and $c'$ is the particle number concentration (no. cm⁻³). The adsorption of polystyrene particles on a solid surface is often an irreversible process, which can be described using the RSA model ($k_d = 0$). According to Shaaf and Talbot[36], the surface blocking function $B(\theta)$ in the RSA model can be expressed as

$$B(\theta) = 1 - 4\theta + \frac{6\sqrt{3}}{\pi}\theta^2 + \left(\frac{40}{\pi\sqrt{3}} - \frac{176}{3\pi^2}\right)\theta^3. \qquad (3)$$

The surface coverage $\theta$ can be represented by

$$\theta = \frac{\Gamma a}{m}, \qquad (4)$$

where $a$ is the occupied area of a single particle (cm²), $m$ is the mass of a single particle [ng], and $\Gamma$ is the adsorbed mass per unit area (ng cm⁻²). Equation (3) is valid for $\theta < 0.3$[36,54]. At higher surface coverage, the kinetics can be approximated with Eq. (5)[54]. Near the jamming limit ($\theta_\infty = 0.547$) $K_0$ is about 8.98.

$$B(\theta) = K_0 (\theta_\infty - \theta)^3 \qquad (5)$$

Noting that $\Gamma(t) = N(t)m$, $c = c'm$ ($c$ in units of ng cm⁻³), and combining Eqs. (2), (3), and (4) we obtain

$$\frac{d\Gamma(t)}{dt} = k_a c \left(1 - 4\Gamma\frac{a}{m} + \frac{6\sqrt{3}}{\pi}\left(\Gamma\frac{a}{m}\right)^2 + \left(\frac{40}{\pi\sqrt{3}} - \frac{176}{3\pi^2}\right)\left(\Gamma\frac{a}{m}\right)^3\right). \qquad (6)$$

The goal was to determine the adsorption coefficient $k_a$ (cm s⁻¹) and the occupied area per nanoplastic particle PS(ø100 nm) $a$ (cm²). Supplementary Fig. 6a shows the frequency change determined by QCM-D for the adsorption of stabile PS(ø100 nm) from a solution with a concentration ($c$) of 0.1 g L⁻¹ (100 000 ng cm⁻³) onto different surfaces. Adsorbed mass was calculated using Eq. (1). All analysis was based on the 5th overtone (25 MHz, $f_0 = 5$ MHz, $n = 5$). The RSA fittings were performed using

Matlab's Curve Fitting Toolbox. Equation 6 was the custom equation. Trust-Region algorithm was used with the following coefficient starting points $k_a c = (a/m) = 0.1$. Lower and upper bounds for both coefficients were 0 and 10. Because QCM-D measurement detects both water and nanoplastic particles adsorbed on the sensor surface, QCM-D was combined with a direct observation method (SEM imaging coupled with image analysis) to obtain the adsorbed dry mass. The amount of adsorbed nanoplastic particles was calculated from SEM images of dry ultrathin films after the adsorption measurements. SEM images were then analyzed as described above, and the number of particles was determined. Since the mass of a single nanoplastic particle was known ($5.26 \times 10^{-7}$ ng) the total dry mass can be calculated ($\triangle m_{IA}$), which equals to the surface mass density $\Gamma$ (ng cm⁻²). To eliminate the influence of water on QCM-D measurement, the QCM-D data $m(t)$ were rescaled utilizing the surface mass density ($\Gamma$) determined from the SEM images ($\Gamma(t) = m(t)$ * ($\Gamma_{max} / m_{max}$)) where max refers to the maximum value determined by each measurement method.

To estimate the amount of coupled water (water strongly interacting with the nanoplastic particles) the total dry mass of particles at the end of the adsorption process (analyzed via image analysis) was first translated into theoretical frequency change ($\triangle f(IA)$) by applying the Eq. (1). Secondly, the true frequency changes ($\triangle f(QCM)$) obtained from the QCM-D measurements were compared with the theoretical frequency changes to reveal the amount of adsorbed water as exemplified in the following calculation:

Example calculation for determining the theoretical frequency response for CNF surface with adsorbed purified 100 nm particles for the 5th overtone and comparing it to the non-normalized frequency response from the QCM-D measurement.

$$\triangle f_5(IA) = -\frac{\triangle m_{IA} * n}{C} = \frac{419 \text{ngcm}^{-2} * 5}{17.7 \text{ngcm}^{-2}\text{Hz}^{-1}} = -118.5 \text{Hz}$$

$$\frac{\triangle f_5(QCM)}{\triangle f_5(IA)} = \frac{-373.5 \text{Hz}}{-118.5 \text{Hz}} = 3.15\ldots \approx 3.2$$

For CNF and RC surfaces and both sparticle types (purified and stabile) the frequency response was a factor of 3 larger for the experimental frequency compared to the calculated frequency from image analysis. This result indicates that 2/3 of the sensed mass by the QCM-D 5th overtone frequency corresponds to the coupled water. For PS ultrathin film, the result was that 3/4 of the sensed mass by QCM-D was coupled water. According to previous studies, the difference between the hydrated and dry mass is typically a factor of 1.5–40[37]. The factor in the case of adsorbing nanoplastic particles (PS(ø100 nm)) is in the lower range since QCM-D is typically used to study protein adsorption, where the proteins can be thought as soft spheres, including water in their structure. In contrast, in the case of PS(ø100 nm) particles they are hard spheres with water only on the surface. Thus, the amount of water that adsorbs along the particles is smaller than for proteins.

## Data availability
The raw data underlying Figs. 1, 2, 3, Supplementary Figs. 2, 3, 6, 7, 8, 9, 13, 14, 15, Table 1 and Supplementary Tables 2–5 are provided as a Source Data file. All relevant data are available from the authors. Source data are provided with this paper.

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

## Acknowledgements

This work has been done as a part of the Academy of Finland's Flagship Programme under Projects No. 318890 and 318891 (Competence Centre for Materials Bioeconomy, FinnCERES) (I.L., T. Lap, T. Loh, C.J., T.T.). S.A. has been funded by the Academy of Finland's postdoctoral project No. 311608/326262 (BoNC-Fi). We thank Katja Pettersson for technical assistance with AFM imaging.

## Author contributions

I.L. performed the QCM-D experiments and SEM imaging for image analysis. I.L. performed the sample preparation and necessary characterizations for all measurements. I.L. contributed to the interpretation of all data and wrote the first version of the manuscript with the co-authors and participated in finalizing it. T. Lap conducted the image analysis and RSA fittings, interpreted RSA data and wrote the respective parts. T. Loh executed the fluorescent experiments with self-standing films. C.J. executed the fluorescent microscopy experiments with hydrogels. S.A. planned and supervised experimental work of I.L., T. Loh, and C.J. S.A. interpreted and handled fluorescent and QCM-D data. S.A. wrote the first version of the manuscript with the co-authors and prepared the figures. T.T. initiated the research concept, contributed to the experimental planning, supervised the interpretation of all data and finalized the manuscript. All authors approved the manuscript.

## Competing interests

The authors declare no competing interests.

**Additional information**

