## [Peer Review File · Nature Communications]

nature portfolio

Peer Review File

Draft OnlyReviewers' comments:

Reviewer #1 (Remarks to the Author):

The manuscript by Leppänen et al. investigated the use of plant-based nanocellulose networks for capturing the colloidal microplastics. The manuscript tackles an interesting topic and shows intensive man-hour efforts. Nevertheless, it is a little bit difficult to read, lacking some clarity in the presentation due to some pressing concerns related with:

(i) the selection of a single family of polymeric particles (PS is an aromatic polymer) as model of nano and microparticles: what is the rationale for selecting PS as the model particles? Are the nanocellulose-based filter systems also efficient for other colloidal microplastics such as PET, PE, PP, PVC? Please elaborate.

(ii) the adsorption tests were done in phosphate buffer with a single type of microplastics, and do not consider that in real environments the nanocellulose-based filter systems will be subjected to different water sources (ranging from wastewaters to the sites where the nano- and microplastics are produced) where different types of colloidal microplastics can be present.

(iii) the choice of regenerated cellulose and PS films as reference materials (lines 108-109): kindly explain the selection of these two reference materials and compare the results with literature.

(iv) the limited comparison of the attained results with the established literature to corroborate the significance of the work in the corresponding field.

OTHER ISSUES

1. Lines 28-29: please add a reference that supports the sentence and elaborate on the debatable nature of the range sizes, particularly in the case of the nanoplastics.
2. Lines 58, 70, 249, 333: Replace “microfluidistic” with microfluidic.
3. Line 443: replace “are presented in Supplementary Fig. 4” with “are presented in Supplementary Fig. 5”, which is the figure showing the AFM data.
4. Lines 425-432: the RC ultrathin films were prepared by desilylation of trimethylsilyl cellulose (TMSC) which is quite different from the dissolution of microcrystalline cellulose with an ionic liquid used to prepare the RC self-standing films (lines 372-377). Kindly explain.
5. Line 445: replace “the films’ chemical nature” with “the films surface wettability”.
6. Line 488: which software was used for the image analysis? This information should be included.
7. Line 519: The authors mention that “The thickness of the ultrathin film is well below 100 nm”, so what are the thickness values of the other self-standing films?

8. Some of the references are missing the volume or page numbers, and not all of them display the DOI. Please check all references.

9. The micrographs presented in the Supplementary Figure 8 are missing the scale bars.

10. Supplementary information, line 149-150: the statement “they are made of different materials, two are colloidal (CNF, TEMPO-CNF), and two are polymeric (RC, PS)” should be rephrased to avoid any misinterpretation because CNF and TEMPO-CNF are also polymeric materials.

In the current state, I do not recommend this paper to be published in Nature Communications.

Reviewer #2 (Remarks to the Author):

This paper assessed the ability of nanocellulose networks to capture PS micro and nanoplastics and uses an innovative approach to describe the binding mechanisms implicated.

Pollution with micro and nanoplastics in different environments, and particularly in water bodies, is of great interest. The present study is original as it presents the potential of a renewable material (cellulose) for the recovery of these pollutants for quantification and cleaning purposes.

The methodologies used are appropriate and their integration successfully designed. The presentation of data in the figures and tables is clear. However I have some concerns before it could be considered for publication.

1-The study is composed of several experiments in which different combinations of particles and cellulose (and other substrates, PS and RC) have been used. It makes the article difficult to follow, specially taking into account that the M&M section is at the end. I think that each section may present a better (although brief) description of the experiment performed and its objective before presenting the results. The article would also benefit from a synthetic table summarizing the objective and materials used in each experiment.

2-On the other hand the use of the terms micro and/or nanoplastics is not consistent throughout the article. Sometimes only one term is used when in fact we are talking about both micro and nano, e.g. in the title of the article or the first sentence of the abstract..... and all the introduction. That makes unclear whether some discussion points in the article are only applicable to one size class or both.

3-Although the methodology used in this study is innovative it is difficult to evaluate the significance of the results. While the article claims for the interest to use nanocellulose to recover micro and nanoplastics from different environments, including wastewaters, all experiments have been performed in aqueous suspensions. How selective are the nanocellulosic materials to micro and nanoplastics? What would happen in a more realistic suspension containing, for example, colloidal organic matter? Can the presence of other colloids saturate the nanocellulose decreasing its ability to trap the plastic particles? Can other environmental parameters influence the binding process (pH, ionic strength...). I miss some experiments testing more realistic conditions or, at least, a critical discussion regarding this issue. If not, the paper should put more emphasis on the importance of the mechanisms described.

Draft Only

Response to the comments:

As requested by both of the Reviewers and the Editor, we have added data related to different environmental conditions the particles might be subjected to. We have revised the manuscript based on the new evidence accordingly. We also provide new data to demonstrate that the nanocellulose-based capturing process is not restricted to only a single type of microplastics. In this reply, we have responded point by point to the comments on the manuscript presented by the reviewers.

A point-by-point response to the comments by the reviewers:

Reviewer 1

The manuscript by Leppänen et al. investigated the use of plant-based nanocellulose networks for capturing the colloidal microplastics. The manuscript tackles an interesting topic and shows intensive man-hour efforts. Nevertheless, it is a little bit difficult to read, lacking some clarity in the presentation due to some pressing concerns related with:

Comment #1:

The selection of a single family of polymeric particles (PS is an aromatic polymer) as model of nano and microparticles: what is the rationale for selecting PS as the model particles? Are the nanocellulose-based filter systems also efficient for other colloidal microplastics such as PET, PE, PP, PVC? Please elaborate.

Authors:

This is a crucial comment, and we have carried out additional experiments with polyethylene (PE) microplastic particles. We have revised the manuscript and Supplementary Information accordingly to correspond our improved understanding on the matter. In the following, we elaborate the selection of the particle systems.

A recognized challenge with nanoplastics is that we lack reliable analytical tools to assess their abundance, source, and fate in the environment, and thus, we lack knowledge on their impact. We need reliable analytical methodology for detecting, identifying, and quantifying these particles, and we need efficient materials and approaches to collect the particles. Our manuscript addresses the both primary needs: (i) how to collect and bind even the most challenging nano- and microplastic fractions from the aqueous environment, and (ii) how to quantitatively analyze those. We have selected the well-defined model system based on polystyrene (PS) particles ($\phi = 100$ nm and $\phi = 1.0$ μ m) with homogeneous size

distribution, which represent the submicron, colloidal nanoplastics i.e. the most challenging and least understood fraction with respect to sampling and identification.

With the aid of the well-defined model system, we are able to systematically reveal the mechanisms enabling the development of efficient technologies to separate, recover, sample, and identify nano- and microplastics from aqueous environment. In this context, we primarily use PS particles since our aim is to develop methods to capture and analyze the colloidal plastic fraction. Thus, the particle size and size distribution need to be uniform and explicit, which is the case with PS particles. Therefore, aqueous PS dispersions serve as an accurate model system for colloidal plastic particles that can be reliably exploited when the aim is to bridge the methodological gap (as defined by Schwaferts *et al.* 2019) on sampling and identification of submicron plastic particles.

According to the report on global assessment of microplastics, the most common microplastics in the environment are PS, PET, PP, and PVC (GESAMP, 2015). PS is the only commercially available and relevant polymeric particle system that qualifies the demand of environmental significance along with the explicit physicochemical character that warrants our approach. In addition, PS particles are generally used in investigations where nano- and microparticle uptake by organisms are studied. In literature, PS systems are often utilised as model systems for nanoplastics. This is probably due to the lack of other polymeric particles available with such uniform size distribution especially in the nanoscale size region.

The applicability of nanocellulose networks in the presence of other nano- and microplastics is a relevant question, which we attempt to elaborate using polyethylene (PE) particles although their properties are much more irregular when compared with PS systems. Despite of the dissimilarities with respect to size and dispersion behaviour (the stability of PE systems is detergent assisted), we provide data showing that self-standing nanocellulose films can entrap PE particles, which display the size range of ϕ 38 - 45 μm . We quantified the entrapped mass of PE particles by nanocellulose films, and we were able to compare the area of nanocellulose film needed to collect all particles evidencing that ability of nanocellulose to capture nanoplastics and microplastics is not limited to a single polymer family. The new data is provided in Supplementary Information (Supplementary Table 4) with relevant descriptions added to the manuscript (page 8). The Methods section is updated accordingly related to PE particles and their characteristics (page 18-19 and 23).

The advanced surface sensitive methods provide explicit information on the material interfaces and surface interactions. Indeed, phenomena related to nanoscale particles mostly occur at interfaces strongly justifying the use of surface sensitive methods for tackling the nanoplastic challenge. In this manuscript, we introduce a surface sensitive method, particle adsorption monitoring by QCM-D, to reveal the role of surface interactions on nanoplastic uptake and binding. The work is again primarily carried out with the well-defined colloidal PS system in order to generate systematic data for more thorough analysis on surface coverage and adsorption kinetics. However, we wanted to make an effort to elaborate the role of surface interactions in PE systems as well. Since PE particles (or other colloidal polymeric systems) are only available as heterogeneous dispersions of nano- and microparticles, we filtrated the PE dispersion to gain a nanoparticle fraction (ϕ 200 - 9900 nm \rightarrow ϕ <450 nm) for the QCM-D adsorption tests. The results showed that the PE nanoplastic fraction did not adsorb on nanocellulose surface although self-standing nanocellulose films were shown to be able to capture PE based microplastics. We provide new

PE adsorption data in Supplementary Information (Supplementary Fig. 9) and we have revised the manuscript (page 10-11) accordingly stating that the hygroscopic feature of nanocellulose and water transport functions taking place in the nanocellulose network are mainly dictating the microplastic capturing process. The Methods section is updated to describe PE materials including fractionation and QCM-D measurements (page 18-19 and 27).

Finally, we would like to highlight an important aspect related to microplastics, especially nano- and colloidal sized particles, and their appearance in nature. They tend to accumulate a stabilizing layer of molecules on their surface. This type of accumulation has been shown by for example toxins' accumulating on the microplastic particle surface (Zhang *et al.*, 2017). The accumulation of molecules on the particle surface masks the chemical characteristics and identity of the specific polymer type, and makes the polymer type of the particles in many ways a secondary question for environmental samples. Our detergent stabilized particles (both PS and PE) resemble or model this type of environmentally stabilized particles. Therefore, we believe that the selected nano and microplastic systems in this work are presenting an appropriate range and variation of particles to be investigated in order to increase the mechanistic understanding to develop efficient capturing materials for nano- and microplastic recovery purposes.

Schwaferts, C., Niessner, R., Elsner, M. & Ivleva, N. P. Methods for the analysis of submicrometer- and nanoplastic particles in the environment. *TrAC - Trends in Analytical Chemistry* vol. 112 52–65 (2019).

GESAMP, 2015. Sources, fate and effects of microplastics in the marine environment: a global assessment. Kershaw, P. J., ed. GESAMP No. 90, 96 p.

Zhang, C., Chen, X., Wang, J. & Tan, L. Toxic effects of microplastic on marine microalgae *Skeletonema costatum*: Interactions between microplastic and algae. *Environ. Pollut.* 220, 1282–1288 (2017).

Comment #2:

The adsorption tests were done in phosphate buffer with a single type of microplastics, and do not consider that in real environments the nanocellulose-based filter systems will be subjected to different water sources (ranging from wastewaters to the sites where the nano- and microplastics are produced) where different types of colloidal microplastics can be present.

Authors:

This is a relevant comment and we have carried out additional adsorption experiments with QCM-D using submicron polystyrene particles in environments simulating *e.g.* waste and laundry waters. This was carried out by changing pH (pH 8 and pH 10) and ionic strength (40 mM NaCl and 200 mM NaCl,) and comparing the nanoplastic adsorption results to the adsorption data achieved in model conditions (pH 6.8, 10 mM phosphate buffer).

Model conditions (phosphate buffer pH 6.8, 10 mM) represent conditions where pH is neutral and the influence of electrolyte addition is negligible to screen the electrostatic interactions. Investigations in

model conditions were carried out with both stable and purified polystyrene (PS) particles, which resemble two types of particles from surface chemistry perspective, *i.e.* fully detergent stabilized PS particles (in the manuscript referred to as stable PS particles) and purified PS particles carrying only a weak surface stabilizing shell (in the manuscript referred to as purified PS particles). Based on the data achieved from the well-defined model conditions we can expand our investigations, and we have now illuminated the influence of environmental parameters on particle adsorption behavior by systematically investigating nanoplastic binding using stable PS particles at different pH levels (Supplementary Fig. 7) and NaCl concentrations (Supplementary Fig. 8).

The pH threshold values for wastewaters in Finland are between pH 6-11. However, wastewaters generally have a pH near neutral due to neutralization. In addition, the pH of wastewater is dependent on the level of purification (entry to water treatment plant to the final stages of purification). We envision that the nanocellulose-based capturing material would function close to the end of the wastewater purification line making the original experiments done in model conditions (pH 6.8 and 10 mM) also relevant from practical point of view. Raw laundry detergents tend to have higher pH values (pH 9-12). However, in use they are strongly diluted with water, which decreases their pH near the pH of tap water (pH 7.3-8.4).

The results from the pH and electrolyte experiments showed that *i)* change in pH does not significantly affect the nanoplastic adsorption, and *ii)* once electrostatic repulsion is screened, the nanocellulose surfaces are showing improved capacity to remove colloidal nanoplastics. We provide new PS adsorption data as a function of pH (Supplementary Fig. 7) and NaCl concentration (Supplementary Fig. 8) in the Supplementary Information, and we have revised the manuscript accordingly (page 10-11). In addition, the Methods section is updated to describe PS adsorption experiments on CNF, TEMPO-CNF and RC with QCM-D (page 27). These fresh results show that well-defined systems involving both nanoplastic particles and model environment are of high importance in order to develop and introduce novel analytical methods to monitor nanoplastic capturing, and to achieve increased understanding on the underlying mechanisms also in different environmental conditions.

We clarify and provide thorough explanation on the choice of particle systems (PS and PE) in Comment#1 by Reviewer 1.

Comment #3:

The choice of regenerated cellulose and PS films as reference materials (lines 108-109): kindly explain the selection of these two reference materials and compare the results with literature.

Authors:

We have added the following explanation to the manuscript to clarify the choice of the reference materials (page 6): “The polymeric regenerated cellulose (RC) and hydrophobic polystyrene (PS) films were used as reference materials to elaborate the influence of morphology and hydrophobic/hydrophilic balance on the capturing process. CNF grades represent fibrillar nanoporous cellulosic structures whereas RC with comparable surface wetting behavior is polymeric without distinguishable porosity

(Supplementary Fig. 5). Furthermore, PS surface is assumed to attract PS particles due to hydrophobic effect and similar chemical structure hereby providing a viable reference for CNF.”

We are creating novel approaches for quantitative analysis of nano- and microplastics from aqueous dispersions, and perfect reference materials do not necessarily exist and, therefore, we have to be able to determine and justify the reference materials by our own experience. PS particle adsorption on regenerated cellulose (RC) has been published by Bhattacharya *et al.* (Bhattacharya *et al.*, 2010). The purpose of the authors was to study the adsorption of nanoplastic particles on to a model system representing the cell wall (cellulose is a major component of many cell walls) and the fate of plastic particles in aquatic systems. In contrast to us, the authors concluded that the medium affinity of RC towards PS nanoparticles was not dependent on solvent salinity. We added this article to the reference list (Reference #31) and we cite it when we discuss the choice of relevant reference materials (page 6).

We have also cited an article where the ability of seagrass to naturally trap plastic waste in aquatic environment has been studied. This study calculates the amount and size distribution of plastic particles trapped in the seagrasses (Sanchez-Vidal *et al.*, 2021). We have compared our materials to the seagrass capturing ability (page 8).

Bhattacharya, P., Chen, R., Lard, M., Lin, S. & Ke, P. C. Binding of nanoplastics onto a cellulose film. INEC 2010 - 2010 3rd Int. Nanoelectron. Conf. Proc. 803–804 (2010) doi:10.1109/INEC.2010.5425197.

Sanchez-Vidal, A., Canals, M., de Haan, W. P., Romero, J. & Veny, M. Seagrasses provide a novel ecosystem service by trapping marine plastics. *Sci. Rep.* 11, 254 (2021).

Comment #4:

The limited comparison of the attained results with the established literature to corroborate the significance of the work in the corresponding field.

Authors:

Studies that are evidencing the capturing and analysis of nanoplastic particles in model or environmental systems are scarce, which makes the direct comparison with the existing systems challenging. In general, the nanoplastics recovery and recognition is an emerging research field where multidisciplinary efforts of biological, physical, chemical, materials and environmental sciences are needed to mitigate the microplastic challenge. The methodological gap in nanoplastic accumulation, production, types, fate and qualitative and quantitative analysis from real samples has been only recently recognised by Schwaferts *et al.* 2019, Wang *et al.* 2021 and Fu *et al.* 2020. In this context, our work showcases the potential of nanocellulose to capture nano- and microplastic particles, and we demonstrate the potential of surface sensitive methods as novel analysis tools to quantify the capturing process. In the Introduction section of our manuscript, we have discussed the issue on existing microplastic capturing methods that rely on size to be insufficient for nanoparticles as they tend to escape the conventional methods (page 3).

The following references are cited in the Introduction:

Schwaferts, C., Niessner, R., Elsner, M. & Ivleva, N. P. Methods for the analysis of submicrometer- and nanoplastic particles in the environment. *TrAC - Trends in Analytical Chemistry* vol. 112 52–65 (2019).

Wang, L. et al. Environmental fate, toxicity and risk management strategies of nanoplastics in the environment: Current status and future perspectives. *J. Hazard. Mater.* 401, (2021).

Fu, W., Min, J., Jiang, W., Li, Y. & Zhang, W. Separation, characterization and identification of microplastics and nanoplastics in the environment. *Sci. Total Environ.* 721, 137561. (2020)

Other issues:

1. Lines 28-29: please add a reference that supports the sentence and elaborate on the debatable nature of the range sizes, particularly in the case of the nanoplastics.

Authors:

References have been changed and updated regarding the primary and secondary micoplastics:

- Schwaferts, C., Niessner, R., Elsner, M. & Ivleva, N. P. Methods for the analysis of submicrometer- and nanoplastic particles in the environment. *TrAC - Trends in Analytical Chemistry* vol. 112 52–65 (2019). (Reference #5)
- GESAMP. Sources, Fate and Effects of Microplastics in the Marine Environment : Part 2 of a global assessment. <https://archimer.ifremer.fr/doc/00632/74391/> (2016). (Reference #6)

In addition, discussion related to the debatable nature of the different size ranges and terminology has been added to the Introduction with additional references (page 2).

- Nanoplastic should be better understood. *Nat. Nanotechnol.* 14, 299 (2019). <https://doi.org/10.1038/s41565-019-0437-7>. (Reference #7)
- Wang, L. et al. Environmental fate, toxicity and risk management strategies of nanoplastics in the environment: Current status and future perspectives. *J. Hazard. Mater.* 401, (2021). (Reference #8)
- Gigault, J. et al. Current opinion: What is a nanoplastic? *Environ. Pollut.* 235, 1030–1034 (2018). (Reference #9)
- (2011/696/EU) Commission Recommendation on the Definition of Nanomaterial European Commission. (2011). (Reference #10)

2. Lines 58, 70, 249, 333: Replace “microfluidistic” with microfluidic.

Authors:

“Microfluidistic” replaced with microfluidic. (Lines 66, 84, 111, 419 in revised manuscript)

3. Line 443: replace “are presented in Supplementary Fig. 4” with “are presented in Supplementary Fig. 5”, which is the figure showing the AFM data.

Authors:

Replaced to Supplementary Fig. 5. (Line 539 in revised manuscript)

4. Lines 425-432: the RC ultrathin films were prepared by desilylation of trimethylsilyl cellulose (TMSC) which is quite different from the dissolution of microcrystalline cellulose with an ionic liquid used to prepare the RC self-standing films (lines 372-377). Kindly explain.

Authors:

The surface and material yielding from both processes are cellulose II. One method is for cellulose II ultrathin film preparation, which cannot be done from ionic-liquid dissolved sample. The method for preparing the self-standing RC films is an established method for the preparation of regenerated cellulose i.e. cellulose II self-standing films.

5. Line 445: replace “the films’ chemical nature” with “the films surface wettability”.

Authors:

Replaced to films’ surface wettability (Line 541 in revised manuscript).

6. Line 488: which software was used for the image analysis? This information should be included.

Authors:

We used Matlab for SEM image analysis and the Analysis Studio 3.11 for the AFM images. Both added to Materials and Methods (Lines 600 and 538 respectively in revised manuscript)

7. Line 519: The authors mention that “The thickness of the ultrathin film is well below 100 nm”, so what are the thickness values of the other self-standing films?

Authors:

The self-standing films’ thicknesses are between 17-230 μm , TEMPO-CNF being the thinnest and polystyrene the thickest.

8. Some of the references are missing the volume or page numbers, and not all of them display the DOI. Please check all references.

Authors:

References have been checked and updated to contain volume and page numbers. The manuscript uses the *Nature* referencing style. The referencing style does not contain the DOI for journal articles. Only conference abstracts, numbered patents and research dataset are refererred with the DOI.

- Example of *Nature* referencing style from Nature submission guidelines: Eigler, D. M. & Schweizer, E. K. Positioning single atoms with a scanning tunnelling microscope. *Nature* **344**, 524-526 (1990).
- Submission guidelines: “Published conference abstracts, numbered patents and research datasets that have been assigned a digital object identifier (DOI) may be included in the reference list.”

9. The micrographs presented in the Supplementary Figure 8 are missing the scale bars.

Authors:

The images in Supplementary Fig. 8 are cropped and zoomed images from Supplementary Fig. 7. Scale bars (500 nm) have been added to the images (Supplementary Fig. 11 in revised Supplementary Information).

10. Supplementary information, line 149-150: the statement “they are made of different materials, two are colloidal (CNF, TEMPO-CNF), and two are polymeric (RC, PS)” should be rephrased to avoid any misinterpretation because CNF and TEMPO-CNF are also polymeric materials.

Authors:

Yes, they are colloids formed by polymers but as materials they do not behave as polymeric materials but as colloids, which is essentially very different. In Supplementary Fig. 5 the difference in structure is clearly visible. We do not see a confusion in the way the materials are described, however, if the colloidal materials (CNF and TEMPO-CNF) would be described as polymeric materials it would confuse the reader a lot more.

Reviewer 2

This paper assessed the ability of nanocellulose networks to capture PS micro and nanoplastics and uses an innovative approach to describe the binding mechanisms implicated.

Pollution with micro and nanoplastics in different environments, and particularly in water bodies, is of great interest. The present study is original as it presents the potential of a renewable material (cellulose) for the recovery of these pollutants for quantification and cleaning purposes.

The methodologies used are appropriate and their integration successfully designed. The presentation of data in the figures and tables is clear. However I have some concerns before it could be considered for publication.

Comment #1:

The study is composed of several experiments in which different combinations of particles and cellulose (and other substrates, PS and RC) have been used. It makes the article difficult to follow, specially taking into account that the M&M section is at the end. I think that each section may present a better (although brief) description of the experiment performed and its objective before presenting the results. The article would also benefit from a synthetic table summarizing the objective and materials used in each experiment.

Authors:

We agree with Reviewer 2 that as several particle/material/experimental set-up combinations have been used the manuscript could benefit from having the Methods section in the beginning. However, the Methods section is at the end as it is usually presented this way in the journal (journal guidelines). To make the manuscript easier to follow, we now present a brief description at the beginning of each section by defining the particles used, the experimental set-up, and the key objective of that experiment. In addition, the idea of a summarizing table was greatly appreciated and we have added such a Table (Table 2) at the end of the manuscript (page 17).

Comment #2:

On the other hand the use of the terms micro and/or nanoplastics is not consistent throughout the article. Sometimes only one term is used when in fact we are talking about both micro and nano, e.g. in the title of the article or the first sentence of the abstract..... and all the introduction. That makes unclear whether some discussion points in the article are only applicable to one size class or both.

Authors:

This is a relevant comment and we see the confusion here as the terms for different size-ranges are still under debate. The term microplastic is generally used for microplastics of all sizes (1 μm -5mm) and there is officially no lower limit to the size of microplastic. Thus, in the previous manuscript version when generally speaking of microplastics (both nano- and micro-sized) we have used the term microplastics. Only in the past few years scientists have started to use the term nanoplastic for particles smaller than a few micrometers. (Nanoplastic should be better understood, 2019) However, this definition, especially the upper limit is still under debate. Some studies have set the upper limit to 1000 nm and others to 100 nm (Gigault *et al.*, 2018). The official size of a nanoparticle for example according to the EU Commission is 1-100 nm (2011/696/EU). In our study, we consider the 100 nm particles as nanoplastic particles and the particles $\geq 1 \mu\text{m}$ as microplastic particles. To make the terminology more clear, we try not to use the term microplastic when dealing with both of these sizes and rather use both micro- and nanoplastic particle. The particles are now also named according to the plastic type (PS or PE) and the size of the spherical particle, e.g. PS(ϕ 100nm), PS(ϕ 1 μm), PE(38-45 μm), PE(200-9900nm) and PE(<450nm) to make it easier to follow.

We focus our study on the polystyrene particles, as they are in the sub μ - and nanoplastic range, 1 nm-1 μm , (Figure below) where traditional purification methods do not apply (Schwaferts *et al.*, 2019). The PE particles (PE(ϕ 38-45 μm) and PE(ϕ 200-9900nm)) are both in the microplastic size range and for them traditional purification methods do apply.

We have added discussion related to the debatable nature of the different size ranges and terminology to the Introduction (page 2).

Figure from Schwaferts *et al.* 2019.

Nanoplastic should be better understood. *Nat. Nanotechnol.* 14, (2019).

Gigault, J. et al. Current opinion: What is a nanoplastic? *Environ. Pollut.* 235, 1030–1034 (2018).

(2011/696/EU), Recommendation on the definition of a nanomaterial.

Schwaferts, C., Niessner, R., Elsner, M. & Ivleva, N. P. Methods for the analysis of submicrometer- and nanoplastic particles in the environment. *TrAC - Trends Anal. Chem.* 112, 52–65 (2019).

Comment #3:

Although the methodology used in this study is innovative it is difficult to evaluate the significance of the results. While the article claims for the interest to use nanocellulose to recover micro and nanoplastics from different environments, including wastewaters, all experiments have been performed in aqueous suspensions. How selective are the nanocellulosic materials to micro and nanoplastics? What would happen in a more realistic suspension containing, for example, colloidal organic matter? Can the presence of other colloids saturate the nanocellulose decreasing its ability to trap the plastic particles? Can other environmental parameters influence the binding process (pH, ionic strength...). I miss some experiments testing more realistic conditions or, at least, a critical discussion regarding this issue. If not, the paper should put more emphasis on the importance of the mechanisms described.

Authors:

In the revised manuscript, we have responded on concerns related to realistic conditions vs. importance of the mechanisms raised by Reviewer 2. We have now performed broad investigations on the influence of environmental parameters on the binding process in order to simulate more realistic conditions.

Furthermore, by carrying out these additional adsorption experiments, we are now able to reveal the capturing mechanism more thoroughly. Please see also the response to Comment#2 by Reviewer 1, where we provide a thorough response related to the new experiments in different environmental conditions. In addition, we provide new data and discussion related to the specificity of the nanocellulose materials using also polyethylene (PE) particles. Please see also Comment#3 by Reviewer 1, where we address the issue in more detail. We have also elaborated how the model conditions actually represent real conditions when dealing with certain type of the wastewaters. We envision that the possible product made from nanocellulosic materials would function close to the end of the wastewater purification line making the original experiments done in model conditions (phosphate buffer pH 6.8 and 10 mM) to be relevant as well. With the nanocellulose mesh we can claim that we are able to uptake the tiniest fraction of nanoplastics which escape the existing purification systems based on e.g. filtration and sieving. Therefore, although being an important factor from the operational point of view, we consider the network saturation by other components e.g. organic matter a topic meriting much deeper study taking place when our approach will be optimized based the real application where the wastewaters are better defined.

Draft Only

REVIEWERS' COMMENTS

Reviewer #1 (Remarks to the Author):

The authors have addressed most of the comments of the reviewers, performed the suggested changes and complemented the work with additional data to demonstrate the capturing of colloidal nano- and microplastics by plant-based nanocellulose networks, which substantially improved the manuscript in terms of readability, clarity and scientific soundness. Therefore, the manuscript can be accepted for publication on Nature Communications without further revision.

Reviewer #2 (Remarks to the Author):

The authors have revised the manuscript according to reviewers' comments. I congratulate the authors for the additional experiments performed, they have strongly improved the quality of the manuscript. Therefore, I suggest this paper to be accepted for publication.